# The Role of the Bacterial Community in Producing a Peculiar Smell in Chinese Fermented Sour Soup

**DOI:** 10.3390/microorganisms8091270

**Published:** 2020-08-21

**Authors:** Liangjing Lin, Jinyuan Wu, Xi Chen, Libiao Huang, Xiaoyong Zhang, Xiangyang Gao

**Affiliations:** 1Guangdong Provincial Key Laboratory of Nutraceuticals and Functional Foods, College of Food Science, South China Agricultural University, Guangzhou 510642, China; jing109@stu.scau.edu.cn (L.L.); wjy1713263483@163.com (J.W.); cxjenny123@163.com (X.C.); 15768309718@163.com (L.H.); 2Joint Laboratory of Guangdong Province and Hong Kong Region on Marine Bioresource Conservation and Exploitation, College of Marine Sciences, South China Agricultural University, Guangzhou 510642, China

**Keywords:** sour soup, flavour, bacteria, Chinese fermentation, GC-IMS, high-throughput sequencing

## Abstract

In this paper, the volatile flavour constituents and the bacterial diversity in characteristic Chinese fermented sour soup were analysed, and the dynamics of bacteria associated with the odour were characterized. The bacterial diversity of sour soup was studied by high-throughput sequencing. A total of 10 phyla and 89 genera were detected. Firmicutes was the dominant phylum of sour soup, accounting for 87.14–98.57%. The genus structure of normal sour soup was relatively simple, and *Lactobacillus* (78.05–90.26%) was the dominant genus. In addition to Lactobacillus, the foul-smelling sour soup contained more *Pediococcus* spp., *Caproiciproducens* spp., and *Clostridium-sensu-stricto*
*12* spp. (relative abundance >1%) than the normal sour soup. A total of 51 aroma compounds were detected by gas chromatography-mass spectrometry(GC-IMS), including 25 esters, 8 terpenes, 8 alcohols, 3 sulfur compounds, 2 acids, 2 ketones, 1 pyrazine, 1 monoterpene and 1 aldehyde. According to the relative odour active value (ROAV) calculation, 51 important flavour-contributing substances and 7 flavour-coordinating substances were determined. The esters with the highest relative percentages and ROAV values provided the pleasant flavour of the sour soup. In the foul-smelling sour soup, the ROAV values of 1,8-cineole, isobutyl acetate, ethyl butanoate, ethyl octanoate-M, and ethyl hexanoate-M decreased, while those of diallyl disulfide-M and diallyl disulfide-D, which were probably responsible for the foul flavour, increased. Through Pearson correlation analysis, the odour production of the foul-smelling soup was determined to be related to *Pediococcus* spp., *Caproiciproducens* spp., *Clostridiumsensu_stricto_12* spp., *Oscillibacter* spp., *Bacteroides* spp., *Fibaculaceae_unclassified* spp., *Acinetobacter* spp. and *Halomonas* spp.

## 1. Introduction

Sour soup is a traditional fermented condiment of Dong and Miao nationalities in Guizhou, China [1]. It is usually fermented naturally and can be made into red and white sour soup. Red sour soup is often made from ingredients such as peppers or tomatoes. Kaili Red sour soup is a typical Guizhou sour soup with a fresh, sweet, and spicy flavour. It was awarded the Guizhou Geographical Indication Protection Product in 2013. In recent years, Guizhou sour soup has gradually expanded outside of Guizhou and become one of the three characteristic hot pot ingredients, alongside Chongqing spicy hot pot sauce and Inner Mongolia clear-soup instant lamb hot pot sauce, with a huge market potential [2]. However, the total output value of Guizhou sour soup is only 240 million yuan, and the degree of industrialization of sour soup is relatively low. In addition, turning the paste into a cylinder during production is an important process that can result in the mixture of the salt and slurry. When the operation of turning the paste is insufficient, it will easily cause the sour soup to become stale and foul smelling. Deterioration of sour soup is a common problem in the production process and greatly affects the industrialization process of sour soup [3]. He Yangbo et al. used electronic nose and gas chromatography-mass spectrometry (GC-IMS) technology to analyse and compare the flavour differences between the foul-smelling sour soup and the normal sour soup and speculated that the foul odour of the sour soup originated from butyric acid and aldehydes [4]. Yang Jinjun et al. used an electronic nose and GC-MS to study the flavour difference of sour soup and found that the main flavours of different brands of sour soup were different [5]. Qiqi Wang et al. studied different kinds of sour soup by high-throughput sequencing and found that *Lactobacillus plantarum* was a common bacterium in the fermented sour soup made from different raw materials, and this bacterium played an important role in the flavour of sour soup [6]. There is a close relationship between flavour formation and the microbes involved in fermentation; however, there are few reports on the relationship between the microbes and flavour generation of sour soup, and there is also a lack of research on the origin of the foul odour of sour soup.

At present, high-throughput sequencing and GC-IMS technology have been widely used in microbial and flavour detection of fermented vegetables [7,8]. Li et al. detected the volatile flavour compounds produced during the curing of ginger and found that the content of heptanoic acid and heptanone decreased, while the content of butyral, butanone, and methionine increased, leading to a change in the flavour of pickled ginger [9]. HuipengLiang et al. used high-throughput sequencing to reveal the traditional Chinese fermentation Paocai and obtained useful information [10]. We believe that high-throughput sequencing and GC-IMS analysis may be a good means to explore the bacterial diversity and flavour variations in sour soup.

In this paper, the correlation between the bacteria and the volatile flavour components of sour soup were studied, and the bacteria that produced the foul-smelling substances in the sour soup were identified, laying the foundation for controlling the production of the odour in the sour soup and promoting the development of the sour soup industry.

## 2. Materials and Methods

### 2.1. Sample Collection and Analysis

The samples were provided by Majiang Mingyang Food Co., Ltd., located in Guizhou province, China. The sample manufacturing flow and collection was as follows: (1) Fresh tomatoes were clean and pulped. (2) The tomato pulp was fermented with salt in a fermentation tank with jar that was turned over during the fermentation period to mix the salt evenly [11]. If the salt is not mixed evenly, the sour soup can become putrid with a foul smell. The foul-smelling sour soup was collected as the sample STF, while the normal sour soup was collected as the control group (TF). The samples were sampled in sextuplicate according to the five-point sampling method studied by DU et al. [12]. A sample was prepared by mixing equal amounts of mixture from the five points: the surface, middle and bottom of the four corners and the midpoint of the fermentation tank, and collected in a sterilized sampling bottle, transported on dry ice to the laboratory and stored at −80 ℃ without loss for further analysis.

### 2.2. Illumina MiSeq Sequencing

Under aseptic conditions, 5 mL samples were thoroughly mixed, and a HiPure Soil DNA kit (Magen, Guangzhou China) was used to extract the total DNA of the samples. A Qubit^®^ dsDNA HS Assay Kit (Qubit v. 3.0, Hilden Germany) was used for the determination of DNA concentration [13]. The purified DNA was used for bacterial 16S rRNA amplification. A total of 20–30 ng DNA was used as a template; the PCR primers included the “CCTACGGRRBGCASCAGKVRVGAAT” sequence and “GGACTACNVGGGTWTCTAATCC” sequence and were designed by GENEWIZ Company (GeneWiz, South Plainfield, NJ, USA) to amplify the V3 and V4 variable regions. In addition, an index connector was added to the end of the 16S rDNA product by PCR for NGS sequencing [14]. The PCR parameters were as follows: predenaturation at 94 °C for 3 min × 1, denaturation at 94 °C for 5 s, annealing at 57 °C for 90 s, elongation at 72 °C for 10 s, and final elongation at 72 °C for 5 min for a total of 24 cycles. Amplification system (20 µL): 2.5 µL TransStart Buffer, 2 µL dNTPs, 1 µL × 2 primer, 0.5 µL TransStart Taq DNA, and 20 ng DNA template; ddH_2_O was added to 25 µL. Library concentrations were detected by Qubit v. 3.0 Fluorometer (Invitrogen, Carlsbad, CA, USA). The library was quantified to 10 nM; PE250/FE300 double-end sequencing was performed according to the instructions of the Illumina MiSeq (Illumina, San Diego, CA, USA) instrument, and sequence information was read by MiSeq Control Software (MCS) [15].

### 2.3. Volatile Flavour Component Analysis

The FlavourSpec^®^ system (Gesellschaft fur Analytische Sensorsy stemembH, Dortmond, Germany) was used to analyse the volatile flavour compounds in this study with an automatic sammer (CTC Analytics AG, Zwingen, Switzerland). The column system of FS-SE-54-CB-1, with a length of 15 m and an inner diameter of 0.53 mm, provided by G.A.S. was used in this work. A 1 g sample was weighed and incubated in a 20 mL headspace glass sampling bottle at 40 °C for 15 min. After incubation, a 500 × L headspace sample was automatically injected into the sampler through a syringe heated to 85 °C. Nitrogen was sampled through the FS-SE-54-CB-1 column. The flow rates were 2 mL/min for 2 min, 15 mL/min for 8 min, and 100 mL/min for 10 min. Volatile compounds were characterized by GC×IMS library search software built into the NIST database and IMS database.

### 2.4. Data Analysis

VSEARCH (v. 1.9.6) was used for sequence clustering (the sequence similarity was set to 97%). The 16S rRNA reference database used was Silva 132. Then, the species classification analysis of the OTU representative sequence was carried out using the RDP Classifier (Ribosomal Database Program) Bayesian algorithm. Based on the OTU analysis results, Shannon, Chao1, and other diversity indexes were calculated. IBM SPSS25.0 (IBM, Armonk, NY, USA) and Microsoft Excel (2016) were used for the normalized calculation and *t*-test of volatile flavour substances. IBM SPSS25.0 (IBM, Armonk, NY, USA) was used to calculate the Pearson correlation between bacteria and volatile flavour substances, and Cystoscope software v. 3.6.0 (v. 3.6.0) was used to visualize the network.

### 2.5. Nucleotide Sequence Accession Numbers

In our study, the raw sequences were uploaded into the NCBI sequence read archive (SRA) database under accession number SRR12066355-SRR12066395.

## 3. Results and Discussion

### 3.1. Sequencing and Alpha Diversity Analysis

As shown in Table 1, a total of 421,235 DNA sequences were obtained. After filtering and rejecting unqualified sequences, 359,570 effective bacterial sequences were obtained from 12 samples. Based on a similarity of 97%, the effective sequences were classified as operational taxonomic units (OTUs), and the number of bacterial OTUs on genera level was 170. The goods coverage of all samples was 1, indicating that the sequencing depth was sufficient to reflect the microbial community contained in the samples. Alpha indexes including ACE, Chao1, Shannon, and Simpson were determined. The Shannon index represents the richness and uniformity of microbes, while the Simpson index is the comprehensive evaluation index of bacterial diversity [16]. The Ace index and Chao1 index represent colony richness; the larger the value is, the higher the richness. The Shannon and Simpson indexes refer to a positive correlation with colony diversity. According to the alpha diversity index, the diversity and richness of the foul-smelling sour soup were higher than those of the normal sour soup.

### 3.2. Analysis of the Difference Between Foul-Smelling and Normal Sour Soup

As shown in Figure 1, the result of principal component analysis (PCA) indicated that the contribution rate of the first principal component was 53.36%, the contribution rate of the second principal component was 34.12%, and the cumulative contribution rate was 87.48%. The data can basically include the main original data. The normal sample and foul-smelling sample were completely separated in the first principal component and the second principal component, and the difference was great. The foul-smelling samples were clustered. However, the normal samples TF3 and TF6 showed a great difference from other samples.

### 3.3. The Bacterial Diversity Difference Between Foul-Smelling and Normal Sour Soup

As shown in Figure 2, a total of 10 phyla were detected. Firmicutes accounted for the largest proportion in the normal samples, accounting for approximately 89.77%; firmicutes are widely found in fermented vegetables [17,18,19]. In addition, the relative abundances of Proteobacteria and Bacteroidetes were 5.1% and 3.65%, respectively. The remaining phyla were Actinobacteria, Verrucomicrobia, Epsilonbacteraeota, Spirochaetes, Chloroflexi, Acidobacteria, and Chlamydiae, but the abundances of these phyla did not exceed 1%. The main phylum of lycopene soup was similar to the studies obtained by Qiqi Wang et al., but there are slight differences in the flora with lower abundance [6]. Compared with the normal samples, the relative abundance of Firmicutes increased by 7.78% in the foul-smelling samples, accounting for 97.55%. The relative abundance of Proteobacteria decreased by 3.24%, accounting for 1.86%. Bacteroidetes, Actinobacteria, Verrucomicrobia, and Epsilonbacteraeota were also detected at a relative abundance of less than 1%. Spirochaetes, Chloroflexi, Acidobacteria, Chlamydiae, or Chlamydiae were not detected in the foul-smelling samples. At the genus level, a total of 89 genera were detected, including 80 genera in the normal samples and 54 genera in the foul-smelling samples. The top genera with the highest abundance are shown in Figure 2, while the remaining genera were classified as others. Lactobacillus (85.88%) was the main genus in the normal samples. Lactobacillus is mainly obtained during vegetable fermentation because it is attached to the surface of vegetables. During the fermentation process, Lactobacillus produces a large number of organic acids, which can not only give fermented vegetables a soft sour taste and aroma but also improve nutritional value and inhibit the growth of harmful bacteria [20]. Moreover, *Bacteroides* spp., *Muribaculaceae*_Unclassified spp., and *Acinetobacter* spp. had relative abundances of 1.62%, 1.49%, and 1.08%, respectively. The relative abundance of other genera was not more than 1%. Similarly, Lactobacillus was predominant in the foul-smelling samples, with a relative abundance of 86.52%. However, compared with the normal samples, the relative abundance of other lactic acid bacteria—*Pediococcus* spp.and *Weissella* spp.—increased in the foul-smelling samples. In addition, *Caproiciproducens* spp.and *Clostridium-sensu-stricto-12* spp., which were not found in the normal samples, had a high abundance in the foul-smelling samples, accounting for 3.67% and 1.68%, respectively. *Caproiciproducens* spp.and *Clostridium-sensu-stricto-12* spp. often appear in the fermentation of wine and can produce butyric acid and caproic acid, giving the wine a pungent and rancid creaminess [21,22,23,24].

### 3.4. Analysis of the Characteristic Bacterial Flora of Normal and Foul-Smelling Sour Soup

Linear discriminant analysis (LEfSe) analysis (Figure 3) showed that there was a significant difference in the bacterial community between the normal samples and foul-smelling samples. The relative abundance of Bacteroidetes, Bacteroidia, Bacteroidia, Bacteroidaceae, Bacteroides, Muribaculaceae, Alphaproteobacteria, and Pseudomonadales in the normal samples was significantly higher than that in the foul-smelling samples. Sensdium-stricto-12, Pediococcus, Caproiciproducens, Ruminococcaceae, Clostridiales, Clostridia and Firmicutes were higher in the foul-smelling samples than in the normal samples.

Figure 4 shows the hierarchical relationship from class to genus. The node size corresponds to the relative abundance of taxa. In the normal samples, the relatively abundant classes are Bacteroidia and Alphaproteobacteria. The orders with higher relative abundance were Bacteroidales and Pseudomonadales, while the families with higher relative abundance were Bacteroidaceae and Muribaculaceae. In the abnormal samples, the class with relatively high abundance was Clostridia, the order with a relatively high abundance was Clostridiales, and the families with relatively high abundances were Clostridiaceae and Ruminococcaceae.

### 3.5. The Relative Content Analysis of Normal and Foul-Smelling Sour Soup

Figure 5 (retention time, migration time, and peak strength) shows that there was a difference in volatile organic compounds between the normal samples and foul-smelling samples.

Figure 5 shows that the headspace components of the normal samples and foul-smelling samples were effectively separated, the flavour components were separated within 1000 s, and the migration time was 7.82–7.85 ms. The signal peak was selected and marked on the ion migration spectrum, and component analysis was carried out by Library Search software built into the GC-IMS. Finally, 51 volatile components and 16 unknown components were clearly identified. The relative percentage content was calculated by the area normalization method and is summarized in Table 2. The *t*-test analysis results of volatile components in the normal and foul-smelling samples are listed in Table 2.

A total of 51 volatile compounds were detected, including 25 esters, 8 terpenes, 8 alcohols, 3 sulfur-containing compounds, 2 acids, 2 ketones, 1 pyrazine, 1 monoterpene, and 1 aldehyde.

Different from other soup soups, where the main volatile flavor is acids or alcohols [25,26], esters were the most abundant aroma compounds detected in the sour soup. Esters contain 5–10 carbon atoms and can provide a pleasant fruity aroma. Esters detected in sour soup samples can be broadly classified into acetic esters and fatty acid ethyl esters. The small size and lipophilic properties of acetic esters make it easier for these compounds to diffuse from the cytoplasm into the extracellular media, while the longer hydrocarbon tails of fatty acid ethyl esters reduces their ability to diffuse across the membrane [27]. As a result, acetic esters have a greater influence on flavour and aroma. The acetic ester with the highest content detected was ethyl acetate, which has the aroma of pineapple fruit. Compared with the normal samples, the relative percentage of esters in the foul-smelling samples decreased significantly.

Alcohols (7.44–12.17%) were second only to esters in sour soup. They are produced by oxidative decomposition of fats or reduction in carbonyl compounds. The presence of alcohol compounds usually produces sweet, fruity, alcoholic, balsam, and green flavours and sensations. The higher content of 1-propanol in alcohols results in a floral, fruity, and grass aroma [28], while the higher content of ethanol, which is mainly produced by lactose fermentation and the catabolic metabolism of alanine, plays an important role in the formation of esters [29]. Compared with the normal samples, the relative content of ethanol and 1-propanol in the foul-smelling samples decreased.

Although only two kinds of ketone substances were detected, the relative percentage content was relatively large and generally presented a floral aroma, with stable and lasting properties. Compared with normal samples, the relative percentages of 2-butanone and acetone in the foul-smelling samples were significantly decreased. Aldehydes are mainly generated by the oxidation of unsaturated fatty acids [30]. The relative normal percentage of 3-methylbutanal in the foul-smelling soup was higher than that in the soup, which has a malty (pungent) taste that is unpleasant at high concentrations but becomes fruity and pleasant at low concentrations [31].

In contrast to the study of He Yangbo [4], we did not detect a large number of acids, and a small amount of propionic acid and butanoic acid was detected. Yang Jinjun et al. found that the volatile substances in different brands of red sour soup were different due to the differences in production technology and raw materials [5]. Xu Li et al. found that well-fermented sour soup contained more acids [32]. Similar to the results of this experiment, the relative percentage of the two acid substances in the foul-smelling samples decreased. Eight species of alkenes were detected; these compounds are generally believed to have originated from plants and carry a special plant aroma.

Sulfur compounds are produced by the enzyme methionine-demethanolase, which cleaves bonds between carbon and sulfur. These compounds are described as having a strong garlic taste and are generally considered to be unpleasant odours. However, at low concentrations, these compounds are considered to be ideal sources of aroma in wine [33].

### 3.6. Evaluation of Relative Odour Activity Value (ROAV) on Key Flavour Components of Sour Soup

The relative aroma activity value (ROAV) was used to calculate the contribution value of volatile substances to flavour, and the main components of the flavour contribution of sour soup were further evaluated objectively. After compounds with ROAV values < 0.01 were eliminated, the resulting ROAV value was used to obtain the heat map in Figure 6.

According to ROAV calculations, the volatile substances with ROAV > 1 were esters (ethyl octanoate-M, ethyl octanoate-D, ethyl hexanoate-M, ethyl hexanoate-D, isoamyl acetate-M, isoamyl acetate-D, ethyl 3-methylbutanoate-D, ethyl butanoate, isobutyl acetate), terpenoids (alpha-phellandrene, linalool, 1,8-cineole), aldehydes (3-methylbutanal), and sulfur-containing compounds (diallyl disulfide-M, diallyl disulfide-D), which play an important role in sour soup flavour. Volatile substances with 0.1 < ROAV < 1 included methyl hexanoate-D, ethyl pentanoate-D, isoamyl acetate-M, ethyl propanoate, alpha-pinene, myrcene, diallyl sulfide.

As shown in Figure 6, ethyl hexanoate-D is the dominant volatile substance in the sour soup; ethyl hexanoate-D has a sweet, fruity, and cucumber aroma [34]. Moreover, the ROAV values of 1,8-cineole, isobutyl acetate, ethyl butanoate, ethyl octanoate-M, and ethyl hexanoate-M of the normal samples were higher. After the peculiar smell was produced, the ROAV value of the foul-smelling samples decreased, and the ROAV values of diallyl disulfide-M and diallyl disulfide-D increased, which may be the main reason for the poor flavour of the foul-smelling samples.

### 3.7. The Link Between Bacteria and the Flavour of Sour Soup

Pearson correlation analysis was conducted for the bacteria (top 30) and the main volatile compounds (ROAV > 1), and the network visualization displayed by Cystoscope software is shown in Figure 7. According to Pearson correlation analysis, LAB are positively correlated with ester substances the same as other fermented vegetables [34], and secondly *Pediococcus* spp., *Caproiciproducens* spp., *Clostridiumsensu_sensu_stricto_12* spp., and Oscillibacter were negatively correlated with ethyl octanoate-M, ethyl hexanoate-M, ethyl butanoate, and 2-methylpropanoate. *Bacteroides* spp., *fibaculaceae*_unclassified spp., *Acinetobacter* spp., and *Halomonas* spp. were negatively correlated with 1,8-cineole. There was a positive correlation between Bacteroides and diallyl disulfide-D. *Pediococcus* spp., *Caproiciproducens* spp., *Clostridiumsensu_stricto_12* spp., Oscillibacter, *Bacteroides* spp., *Fibaculaceae_unclassified* spp., *Acinetobacter* spp. and *Halomonas* spp. may be the microbial factors that lead to flavour changes in sour soup.

## 4. Conclusions

In this paper, Illumina MiSeq sequencing and GC-IMS were used to investigate the role of the bacterial community in producing a peculiar smell in Chinese fermented sour soup. A total of 89 bacterial genera and 51 aroma compounds (including 25 esters, 8 terpenes, 8 alcohols, 3 sulfur compounds, 2 acids, 2 ketones, 1 pyrazine, 1 monoterpene and 1 aldehyde) were detected in the sour soup. *Pediococcus* spp., Caproiciproducens, *Clostridiumsensu_stricto_12,* Oscillibacter, *Bacteroides* spp., *Fibaculaceae_unclassified* spp., *Acinetobacter* spp., and *Halomonas* spp. were negatively correlated with ester and terpene compounds and positively correlated with sulfur-containing compounds, which may be the main cause of the flavour change in sour soup. This study provides microbial and metabolic information for optimizing the fermentation quality of sour soup and solving the odour problem.

## Figures and Tables

**Figure 1 microorganisms-08-01270-f001:**
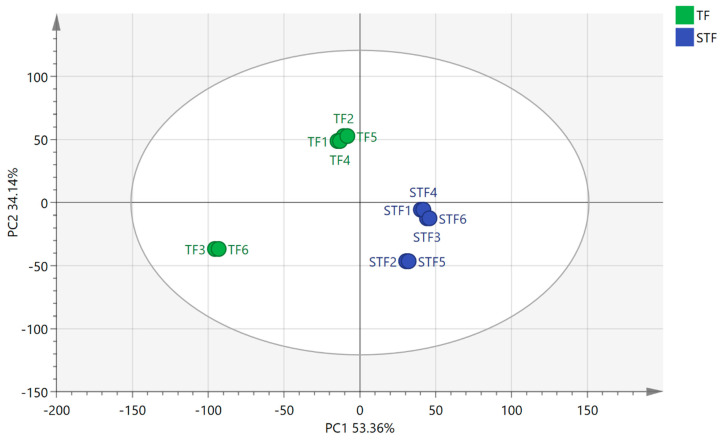
PCA analysis of sour soup samples. TF refers to the normal tomato fermentation sample; STF refers to the foul-smelling tomato fermentation sample; (1–6) indicate a parallel sample of the same group sample.

**Figure 2 microorganisms-08-01270-f002:**
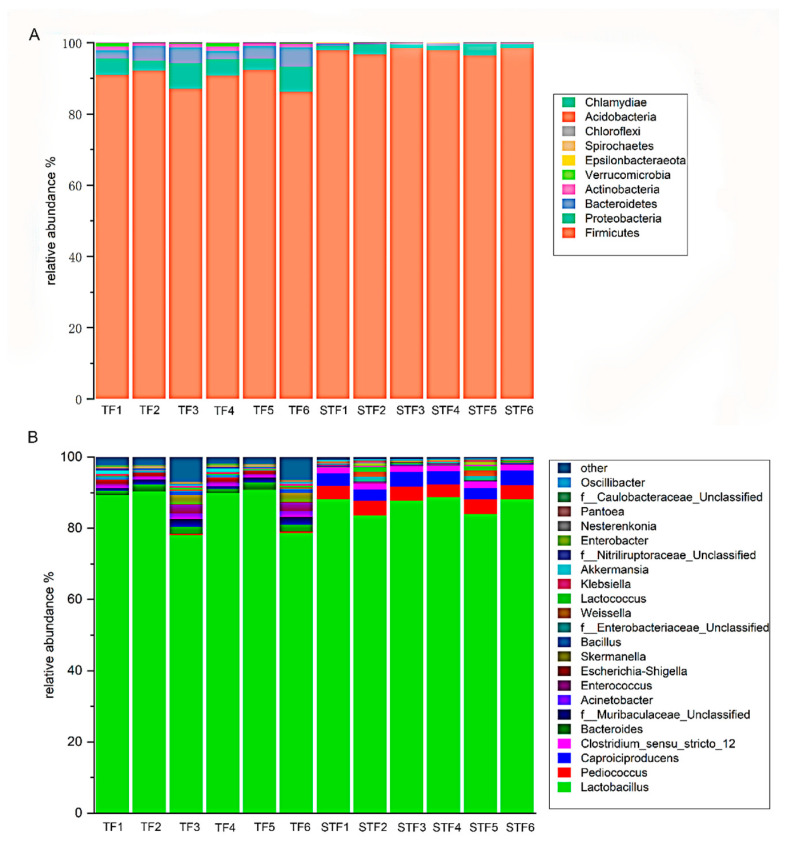
The relative abundance of bacteria in normal and spoilage sour soup at the phylum level (**A**) and genus level (**B**). TF refers to the normal tomato fermentation sample; STF refers to the foul-smelling tomato fermentation sample; (1–6) indicate a parallel sample of the same group sample. Each phylum or genus is represented by a unique colour. Each column represents a different studied sample.

**Figure 3 microorganisms-08-01270-f003:**
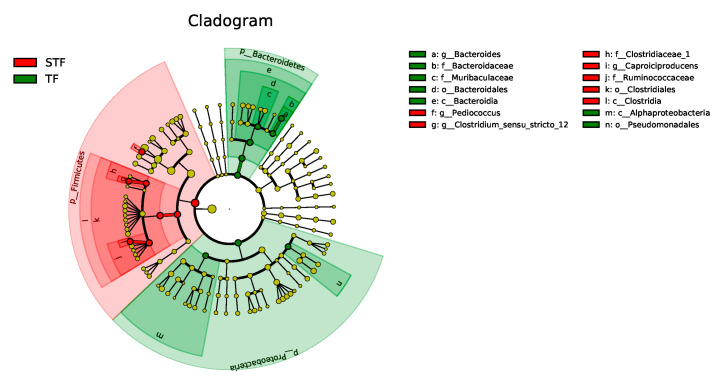
Cladogram of bacterial colonies in sour soup. TF refers to the normal tomato fermentation sample; STF refers to the foul-smelling tomato fermentation sample. The different colours represent the different samples. Node size corresponds to the relative abundance of taxa.

**Figure 4 microorganisms-08-01270-f004:**
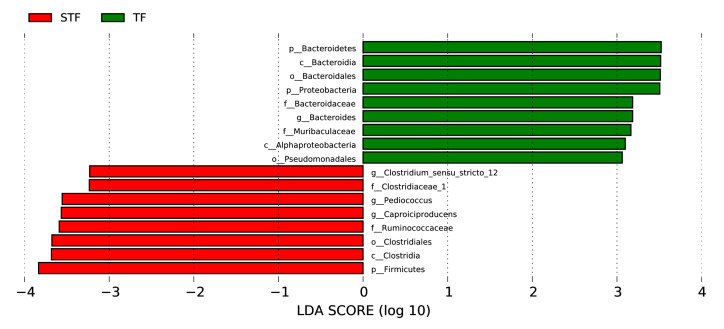
Trilinear discriminant analysis. TF refers to the normal tomato fermentation sample; STF refers to the foul-smelling tomato fermentation sample. The different colours represent the characteristic microorganisms of different samples. Each column represents a different microorganism.

**Figure 5 microorganisms-08-01270-f005:**
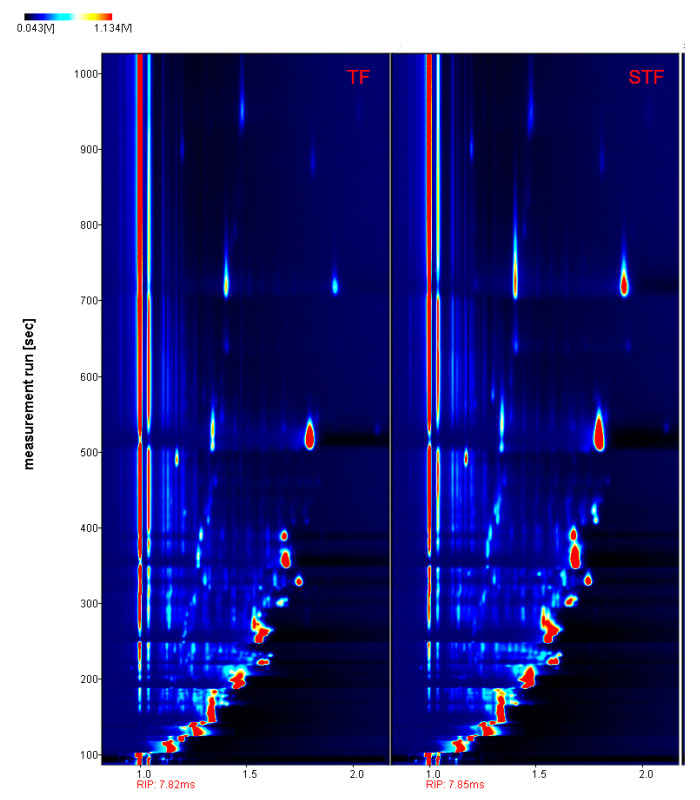
GC-IMS spectra of acid soup (planform). TF refers to the normal tomato fermentation sample; STF refers to the foul-smelling tomato fermentation sample. The red vertical line represents the reactive ion peak (RIP), and each point on both sides of the RIP represents a volatile organic compound. The intensity ranges from deep to light, as indicated, from red to white and from dark to light.

**Figure 6 microorganisms-08-01270-f006:**
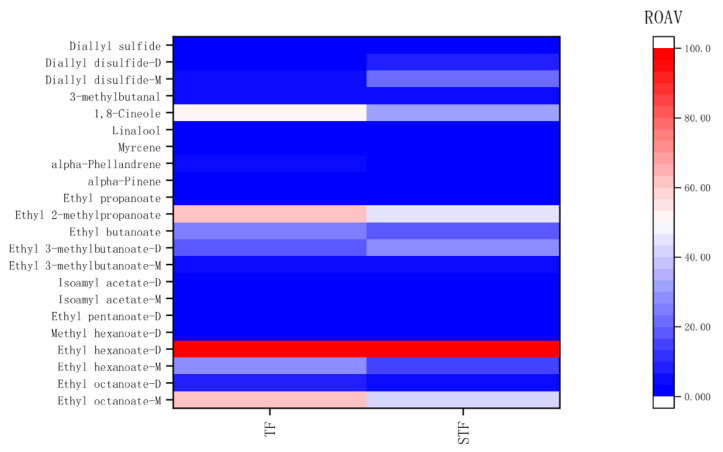
ROAV values of volatile components in sour soup. TF refers to the normal tomato fermentation sample; STF refers to the foul-smelling tomato fermentation sample. The colour code ranges from dark blue (lowest relative abundance) to dark red (highest relative abundance). The colour of each line represents the ROAV value of flavour, and each column represents a different studied sample in the heat map. Datapoints represent the means of three biological repeats. M refers to monomer, and D refers to dimer.

**Figure 7 microorganisms-08-01270-f007:**
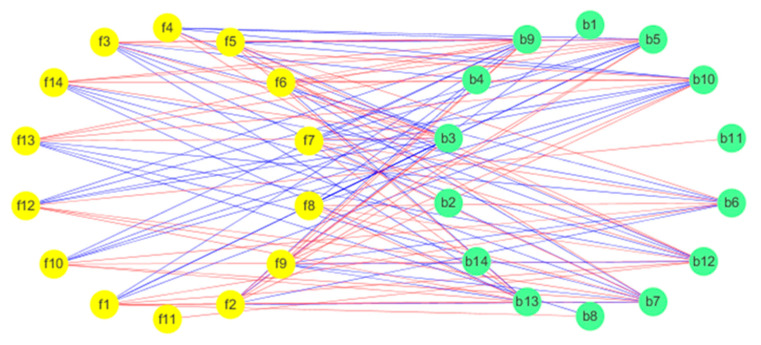
The correlation between microorganisms and volatile flavours. Yellow circles represent flavour components, and green circles represent bacterial genera. f1, ethyl octanoate-M; f2, linalool; f3, isoamyl acetate; f4, ethyl octanoate-D; f5, ethyl hexanoate-M; f6, ethyl hexanoate-D; f7, ethyl butanoate; f8, ethyl 3-methylbutanoate-M; f9, ethyl 3-methylbutanoate-D; f10, ethyl 2-methylpropanoate; f11, diallyl disulfide-D; f12, alpha-phellandrene; f13, 3-methylbutanal; f14, 1,8-cineole. b1, *Bacteroides*; b2, *Ralstonia*; b3, *Pediococcus*; b4, *Parabacteroides*; b5, *Oscillibacter*; b6, *Halomonas*; b7, *f__Muribaculaceae_Unclassified*; b8, *Escherichia-Shigella*; b9, *Clostridium_sensu_stricto_12*; b10, *Caproiciproducens*; b11, *Bifidobacterium*; b12, *Bacteroides*; b13, *Acinetobacter.*

**Table 1 microorganisms-08-01270-t001:** Sequencing data and alpha diversity index.

Sample	Raw Sequence Number	Effective Sequence Number	Goods Coverage	OTUs	Ace	Chao1	Shannon	Simpson
TF1	34041	28881	1	92	92.20	92	1.218	0.233
TF2	46002	27874	1	96	96.00	96	1.122	0.208
TF3	34180	33443	1	128	128.81	128.75	2.28	0.442
TF4	32408	24870	1	91	91.00	91.60	1.18	0.204
TF5	46002	43900	1	94	95.78	94	1.034	0.213
TF6	34560	24673	1	103	124.74	104	2.08	0.235
STF1	30267	28956	1	104	117.5	110.40	1.762	0.462
STF2	30044	33951	1	98	104.2	95.80	2.481	0.529
STF3	29351	23169	1	93	95.9	92.51	1.702	0.419
STF1	30503	26910	1	116	121.74	118.5	1.863	0.479
STF2	47201	36861	1	93	103.09	98.5	2.412	0.582
STF3	26018	26082	1	90	94.31	94	1.877	0.492

TF refers to the normal tomato fermentation sample; STF refers to the foul-smelling tomato fermentation sample; (1–6) indicate a parallel sample of the same group sample.

**Table 2 microorganisms-08-01270-t002:** The relative percentage of volatile components in sour soup.

Classification	Volatile Component	Statistical Significance	Relative Content %	
			TF	STF
Esters	Ethyl octanoate-M	*	1.73 ± 0.02	1.32 ± 0.05
	Ethyl octanoate-D	*	0.26 ± 0.02	0.16 ± 0.03
	Methyl salicylate	*	0.50 ± 0.00	0.59 ± 0.02
	Propyl hexanoate-M	*	3.58 ± 0.07	4.31 ± 0.02
	Propyl hexanoate-D	*	1.64 ± 0.08	5.71 ± 0.01
	Ethyl heptanoate	*	0.37 ± 0.02	0.56 ± 0.01
	Ethyl hexanoate-M	*	3.96 ± 0.03	2.73 ± 0.02
	Ethyl hexanoate-D	*	14.07 ± 0.09	16.43 ± 0.02
	Methyl hexanoate-M	*	0.90 ± 0.02	0.64 ± 0.00
	Methyl hexanoate-D	*	2.64 ± 0.03	3.14 ± 0.01
	Amyl acetate	*	0.25 ± 0.01	0.32 ± 0.00
	Ethyl pentanoate-M	*	1.20 ± 0.00	0.63 ± 0.00
	Ethyl pentanoate-D	*	8.54 ± 0.01	9.97 ± 0.00
	Isoamyl acetate-M	*	0.46 ± 0.00	0.28 ± 0.00
	Isoamyl acetate-D	*	2.58 ± 0.03	2.74 ± 0.00
	Ethyl 3-methylbutanoate-M	*	0.26 ± 0.00	0.22 ± 0.00
	Ethyl 3-methylbutanoate-D	*	1.07 ± 0.00	1.78 ± 0.00
	Ethyl 2-hydroxypropanoate	*	1.55 ± 0.02	1.36 ± 0.00
	Ethyl butanoate	*	6.60 ± 0.07	5.55 ± 0.03
	Isobutyl acetate	*	0.28 ± 0.00	0.40 ± 0.00
	Ethyl 2-methylpropanoate	*	1.70 ± 0.02	1.49 ± 0.02
	Propyl acetate		2.64 ± 0.03	2.65 ± 0.02
	Ethyl propanoate	*	2.46 ± 0.01	1.75 ± 0.03
	Ethyl acetate	*	11.37 ± 0.10	9.37 ± 0.10
	Butyl acetate	*	0.22 ± 0.00	0.24 ± 0.00

Pyrazines	Trimethyl pyrazine		1.36 ± 0.00	1.39 ± 0.00
Acids	Propionic acid	*	0.51 ± 0.04	0.17 ± 0.00
	Butanoic acid	*	0.20 ± 0.04	0.10 ± 0.00
Terpenes	alpha-Pinene	*	0.34 ± 0.00	0.16 ± 0.00
	Limonene		0.14 ± 0.00	0.14 ± 0.00
	3-Carene		0.04 ± 0.00	0.03 ± 0.00
	alpha-Phellandrene	*	0.02 ± 0.00	0.01 ± 0.00
	Myrcene	*	0.07 ± 0.00	0.05 ± 0.00
	beta-Pinene	*	0.04 ± 0.00	0.03 ± 0.00
	Thujene	*	0.08 ± 0.00	0.06 ± 0.00
	Terpinolene	*	0.06 ± 0.00	0.07 ± 0.00
	Linalool	*	0.08 ± 0.00	0.11 ± 0.00

Alcohol	1-Hexanol	*	0.17 ± 0.00	0.09 ± 0.00
	2-Methylbutanol	*	0.46 ± 0.01	0.48 ± 0.00
	1-Propanol	*	4.15 ± 0.03	3.27 ± 0.07
	Ethanol	*	3.96 ± 0.03	3.00 ± 0.06
	(E)-2-hexenol-M	*	0.03 ± 0.00	0.08 ± 0.00
	(E)-2-hexenol-D	*	0.04 ± 0.00	0.22 ± 0.00
	1-Pentanol		0.17 ± 0.00	0.17 ± 0.00

Ketones	2-Butanone	*	2.91 ± 0.02	1.91 ± 0.07
	acetone	*	0.97 ± 0.00	0.47 ± 0.00

Sulfocompounds	Diallyl disulfide-M		0.11 ± 0.01	0.10 ± 0.01
	Diallyl disulfide-D		0.04 ± 0.00	0.03 ± 0.01
	Diallyl sulfide	*	0.13 ± 0.00	0.06 ± 0.00

Monoterpenes	1,8-Cineole	*	0.20 ± 0.03	0.50 ± 0.01
Aldehydes	3-Methylbutanal	*	0.49 ± 0.00	0.54 ± 0.01

TF refers to the normal tomato fermentation sample; STF refers to the foul-smelling tomato fermentation sample. All analyses were conducted in triplicate, and the mean ± SD is presented. M refers to monomer, and D refers to dimer. The * refers to the statistical significance difference between two groups samples.

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
