# Peer review of "The Role of the Bacterial Community in Producing a Peculiar Smell in Chinese Fermented Sour Soup"

_microorganisms, 2020, doi:10.3390/microorganisms8091270_

Round 1
Reviewer 1 Report
In my opinion, it´s a well-organizated article, with all their sections well-developed and a synthesizing literature, easy to understand. The issue of the paper is innovative and original, and the data and analyses are presented appropriately with a wide scientific soundness.
In respect to the correction, I have found only one minor criticisms. In the line 129 I suppose that 1189 bacterial OTUs is refereed to specie level but it could do refer at different taxonomic levels like genus, so I would make it clear.
In general, my assessment is favourable and I agree with the publication of this paper.
Author Response
To Reviewer 1:
Question 1: In my opinion, it´s a well-organizated article, with all their sections well-developed and a synthesizing literature, easy to understand. The issue of the paper is innovative and original, and the data and analyses are presented appropriately with a wide scientific soundness. In respect to the correction, I have found only one minor criticisms.
Response: The authors thank the reviewer for insightful and constructive comments that have been of great help to us in revising the manuscript.
Question 2: In the line 129 I suppose that 1189 bacterial OTUs is refereed to specie level but it could do refer at different taxonomic levels like genus, so I would make it clear.
Response: The following content has been corrected in the part result of the revised manuscript. Please check the revised manuscript. In addition, OTU statistical problems were found during the verification process, which have been corrected.
Based on a similarity of 97%, the effective sequences were classified, and the number of bacterial OTUs on species level was 170.

Reviewer 2 Report
The current report aiming to elucidate the role of the bacterial community in producing a peculiar smell in Chinese fermented sour soup. A total of 10 phyla and 89 genera were detected and Lactobacillus (78.05%~90.26%) was the dominant genus. A total of 51 aroma compounds were detected by GC-IMS, including 25 esters, 8 terpenes, 8 alcohols, 3 sulfur compounds, 2 acids, 2 ketones, 1 pyrazine, 1 monoterpene and 1 aldehyde. The ms explained where the foul-smelling sour soup came from. The findings are important to the audience working in the similar field. In my opinion, the ms can be accepted after minor revision.
- Location of Majiang Mingyang Food Co., Ltd. should be provided.
- 2.4 How did the authors collect and keep their samples without loss?
- More discussion from the results with similar studies are needed.
Author Response
To Reviewer 2:
Question 1: The current report aiming to elucidate the role of the bacterial community in producing a peculiar smell in Chinese fermented sour soup. A total of 10 phyla and 89 genera were detected and Lactobacillus (78.05%~90.26%) was the dominant genus. A total of 51 aroma compounds were detected by GC-IMS, including 25 esters, 8 terpenes, 8 alcohols, 3 sulfur compounds, 2 acids, 2 ketones, 1 pyrazine, 1 monoterpene and 1 aldehyde. The ms explained where the foul-smelling sour soup came from. The findings are important to the audience working in the similar field. In my opinion, the ms can be accepted after minor revision.
Response: The authors are grateful to the review for their valuable comments and concerns, and all comments and concerns are responded carefully point by point.
Question 2: Location of Majiang Mingyang Food Co., Ltd. should be provided.
Response: The following content has been corrected in the part result of the revised manuscript. Please check the revised manuscript.
The samples were provided by Majiang Mingyang Food Co., Ltd located in Guizhou province, China.
Question 2: How did the authors collect and keep their samples without loss?
Response: The following content has been corrected in the part result of the revised manuscript. Please check the revised manuscript.
The samples were sampled in sextuplicate according to the five-point sampling method studied by DU et al [1]. A sample was prepared by mixing equal amounts of mixture from from the 5 points: the surface, middle and bottom of the four corners and the midpoint of the fermentation tank, and collected in sterilized sampling bottle, transported on dry ice to the laboratory and stored at -80℃ without loss for further analysis.
References:
[1] Du, F.-m.; Gu; Song. Dynamic Changes in the Bacterial Community During the Fermentation of Traditional Chinese Fish Sauce (TCFS) and their Correlation with TCFS Quality. Microorganisms 2019, 7, 371, doi:10.3390/microorganisms7090371 %/ MDPI AG.
Question 3: More discussion from the results with similar studies are needed.
Response: The following sentences were added in the revised manuscript to stress the comparison with similar studies.
- The remaining phyla were Actinobacteria, Verrucomicrobia, Epsilonbacteraeota, Spirochaetes, Chloroflexi, Acidobacteria, and Chlamydiae, but the abundances of these phyla did not exceed 1%. The main phylum of lycopene soup were similar with the studies obtained by Qiqi Wang et al.[1], but there are slight differences in the flora with lower abundance. Compared with the normal samples, the relative abundance of Firmicutes increased by 7.78% in the foul-smelling samples, accounting for 97.55%.
- Different from other soup soups, where the main volatile flavor is acids or alcohols, esters were the most abundant aroma compounds detected in the sour soup[2,3].
- According to Pearson correlation analysis, LAB are positively correlated with ester substances the same as other fermented vegetables[4], and secondly Pediococcus, Caproiciproducens, Clostridiumsensu_sensu_stricto_12, and Oscillibacter were negatively correlated with ethyl octanoate-M, ethyl hexanoate-M, ethyl butanoate, and 2-methylpropanoate.
References:
[1] Qiqi, W.; Jiexian, T.; Zongdong, P.; Jing, D. Analysis of dominant microflora associated with the unique flavor of Guizhou Kaili sour Soup using IlIumina MiSeq sequencing. Food and Fermentation Industries 2020, 1-10.
[2] Dong-ya, Z.; Li, X.; Qing-song, L.; Qin, M. Optimization of Kaili red sour soup tomato juice fermentation process and fortified microflora. Food Science and Technology 2018, 43, 313-318.
[3] Sha-sha, Z.; Ping, H.U.; Ping, H.U. Study on Qual ity Change of Red Sour Soup Fermented by Lactic Acid Bacteria. China Condiment 2019, 44, 65-70, doi:10.3969/j.issn.1000-9973.2019.08.012.
[4] Rao, Y.; Tao, Y.; Chen, X.; She, X.; Qian, Y.; Li, Y.; Du, Y.; Xiang, W.; Li, H.; Liu, L., et al. The characteristics and correlation of the microbial communities and flavors in traditionally pickled radishes. LWT 2020, 118, 108804, doi:https://doi.org/10.1016/j.lwt.2019.108804.
